# Clinical Characteristics and In Vitro Analysis of *MYO6* Variants Causing Late-onset Progressive Hearing Loss

**DOI:** 10.3390/genes11030273

**Published:** 2020-03-04

**Authors:** Shin-ichiro Oka, Timothy F. Day, Shin-ya Nishio, Hideaki Moteki, Maiko Miyagawa, Shinya Morita, Shuji Izumi, Tetsuo Ikezono, Satoko Abe, Jun Nakayama, Misako Hyogo, Nobuhiko Okamoto, Natsumi Uehara, Chie Oshikawa, Shin-ichiro Kitajiri, Shin-ichi Usami

**Affiliations:** 1Department of Otorhinolaryngology, Shinshu University School of Medicine, 3-1-1 Asahi, Matsumoto, Nagano 390-8621, Japan; okashin@shinshu-u.ac.jp (S.-i.O.); timothy@shinshu-u.ac.jp (T.F.D.); moteki@shinshu-u.ac.jp (H.M.); miyamai@shinshu-u.ac.jp (M.M.);; 2Department of Hearing Implant Sciences, Shinshu University School of Medicine, 3-1-1 Asahi, Matsumoto, Nagano 390-8621, Japan; 3Department of Otolaryngology-Head and Neck Surgery, Faculty of Medicine and Graduate School of Medicine, Hokkaido University, North-15, West-7, Sapporo 060-8638, Japan; shinyamorita@yahoo.co.jp; 4Department of Otolaryngology, Head and Neck Surgery, Niigata University Graduate School of Medical and Dental Sciences, Asahimachi 1, Niigata city, Niigata 951-8510, Japan; izumi@med.niigata-u.ac.jp; 5Department of Otorhinolaryngology, Saitama Medical University Faculty of Medicine, Morohongo 38, Moroyamamachi, Irumagun, Saitama-ken 350-0495, Japan; ikezono.tetsuo@1972.saitama-med.ac.jp; 6Department of Otorhinolaryngology, Toranomon Hosipital, 2-2-2 Toranomon, Minato-ku, Tokyo 105-8470, Japan; abe3387@ybb.ne.jp; 7Department of Otorhinolaryngology, Shiga University School of Medical Science, Seta Tsukinowacho, Otsu 520-2192, Japan; jfukui@belle.shiga-med.ac.jp; 8Department of Otolaryngology-Head and Neck Surgery, Kyoto Prefectural University of Medicine, 465 Kagii-cho, Kyoto 602-8566, Japan; mihyogo@koto.kpu-m.ac.jp; 9Department of Medical Genetics, Osaka Women’s and Children’s Hospital, 840 Murodo-cho, Izumi, Osaka 594-1101, Japan; okamoto@osaka.email.ne.jp; 10Department of Otolaryngology-Head and Neck Surgery, Kobe University School of Medicine, 7-5-2 Kusunoki-cho, Chuo-ku, Kobe 650-0017, Japan; natsuko0522@gmail.com; 11Department of Otorhinolaryngology-Head and Neck Surgery, Graduate School of Medical Sciences, Kyushu University, 3-1-1, Maidashi, Higashi-ku, Fukuoka 812-8582, Japan; chieoshika@yahoo.co.jp

**Keywords:** *MYO6*, myosin, non-syndromic hearing loss, DFNA22, DFNB37, autosomal dominant, prevalence, genotype–phenotype correlation, hearing progression

## Abstract

*MYO6* is known as a genetic cause of autosomal dominant and autosomal recessive inherited hearing loss. In this study, to clarify the frequency and clinical characteristics of hearing loss caused by *MYO6* gene mutations, a large-scale genetic analysis of Japanese patients with hearing loss was performed. By means of massively parallel DNA sequencing (MPS) using next-generation sequencing for 8074 Japanese families, we found 27 *MYO6* variants in 33 families, 22 of which are novel. In total, 2.40% of autosomal dominant sensorineural hearing loss (ADSNHL) in families in this study (32 out of 1336) was found to be caused by *MYO6* mutations. The present study clarified that most cases showed juvenile-onset progressive hearing loss and their hearing deteriorated markedly after 40 years of age. The estimated hearing deterioration was found to be 0.57 dB per year; when restricted to change after 40 years of age, the deterioration speed was accelerated to 1.07 dB per year. To obtain supportive evidence for pathogenicity, variants identified in the patients were introduced to *MYO6* cDNA by site-directed mutagenesis and overexpressed in epithelial cells. They were then assessed for their effects on espin1-induced microvilli formation. Cells with wildtype myosin 6 and espin1 co-expressed created long microvilli, while co-expression with mutant constructs resulted in severely shortened microvilli. In conclusion, the present data clearly showed that *MYO6* is one of the genes to keep in mind with regard to ADSNHL, and the molecular characteristics of the identified gene variants suggest that a possible pathology seems to result from malformed stereocilia of the cochlear hair cells.

## 1. Introduction

Genetic etiology is the most common cause of sensorineural hearing loss (SNHL), with over 50% of congenital hearing loss attributed to genetic causes in the US [1], a figure that is likely similar in all industrialized nations. More than 100 genes have been reported to be associated with non-syndromic hearing loss [2] (hereditary hearing loss home page https://hereditaryhearingloss.org/). Among them, several myosin genes are known to cause both syndromic and non-syndromic hearing loss (*MYO7A*, DFNA11 [3]; DFNB2 [4]; Usher syndrome type 1B [5]; *MYH9*, DFNA17 [6]; MYH9-related disorder [7]; *MYH14*, DFNA4 [8]; peripheral neuropathy, myopathy, hoarseness, and hearing loss [9]; *MYO6*, DFNA22 [10], DFNB37 [11]; *MYO3A*, DFNB30 [12]; *MYO15A*, DFNB3 [13]).

The human *MYO6* gene is located on chromosome 6q13 and its gene mutations can cause either autosomal dominant (AD) inherited non-syndromic hearing loss (DFNA22) or autosomal recessive (AR) inherited non-syndromic hearing loss (DFNB37), possibly reflecting differences in pathophysiology among the mutations. The *MYO6* gene encodes myosin 6, which is known to play a crucial role as a motor protein for cargo transportation [14]. In the mouse cochlea, myosin 6 proteins are expressed in outer and inner hair cells, particularly in the basal regions of the stereocilia [15,16]. Interestingly, it is also known that only type 6 myosin among the myosin family moves towards the basal end of stereocilia rootlets [17,18]. Seki et al. showed that the stereocilia of *MYO6* mutant mice degenerated gradually and formed fused or branched stereocilia [19]. Therefore, myosin 6 is likely to be involved in the maintenance of stereocilia.

Despite its importance in the inner ear, a limited number of *MYO6*-associated hearing loss cases have been published. We have recently reported seven families with *MYO6* mutations and showed their clinical features [20].

In this study, to obtain a more complete picture of the mutational spectrum, frequency of mutations, and characteristic clinical features of the patients, we undertook an analysis using a larger (*n* = 8074) cohort. In addition, we also performed in vitro functional analysis of novel missense *MYO6* mutations identified in this study to clarify the cellular effects of the mutations.

## 2. Materials and Methods 

### 2.1. Subjects

A total of 8074 Japanese hearing loss patients (AD, 1336; AR or sporadic, 5564; unknown, 1174) from 67 otolaryngology departments nationwide participated in this study. Our previous reported subjects (*n* = 1120) were included in the 8074 subjects [20]. Written informed consent was obtained from all subjects (or from their next of kin, caretaker, or guardian in the case of minors/children) prior to enrollment in the project. This study was approved by the Shinshu University Ethical Committee as well as the respective ethical committees of the other participating institutions listed hereinafter: Niigata University Ethical Committee, Toranomon Hospital Ethical Committee, Shiga University Ethical Committee, Kyoto Prefectural University Ethical Committee, Kyushu University Ethical Committee, Saitama Medical University Ethical Committee, Kobe University Ethical Committee, Osaka Medical Center and Research Institute for Maternal and Children Health Ethical Committee, and Hokkaido University Ethical Committee. All methods were performed in accordance with the Guidelines for Genetic Tests and Diagnoses in Medical Practice of the Japanese Association of Medical Sciences and the Declaration of Helsinki as required by Shinshu University, and the protocol was approved by the Ethics Committee of Shinshu University School of Medicine (No. 387—4 September 2012 and No. 576—2 May 2017).

### 2.2. Clinical Evaluation

Clinical data, including the onset age, progressiveness of hearing loss, pedigree, and episodes of vertigo, were collected based on their anamnestic records. Pure-tone audiometry was performed on patients over the age of 5. For very young children under the aged 4 or under, the auditory steady state response (ASSR), conditioned orientation response audiometry (COR) or play audiometry, was performed. The pure-tone average (PTA) was calculated from the audiometric thresholds at four frequencies (0.5, 1, 2, and 4 kHz). Severity of hearing loss was categorized into four groups: mild (PTA 20–40 dB hearing level (HL)), moderate (41–70 dBHL), severe (71–90 dBHL), and profound (>91 dBHL) [21].

### 2.3. Amplicon Library Preparation 

Amplicon libraries were prepared using an Ion AmpliSeq Custom Panel (Thermofisher Scientific, Waltham, MA, USA), in accordance with the manufacturer’s instructions, for 63 genes reported to cause non-syndromic hearing loss [22]. After preparation, emulsion PCR and sequencing were performed according to the manufacturer’s instructions. The detailed protocol has been described elsewhere [22]. In brief, massively parallel DNA sequencing (MPS) was performed with an Ion Torrent Personal Genome Machine (PGM) system using an Ion PGM 200 Sequencing Kit and an Ion 318 Chip (Thermofisher Scientific, Waltham, MA, USA) or an Ion Proton system with an Ion PI Chip and Ion Hi-Q Chef Kit (Thermofisher Scientific, Waltham, MA, USA).

The sequence data were mapped against the human genome sequence (build GRCh37/hg19) with a Torrent Mapping Alignment Program. After sequence mapping, the DNA variant regions were piled up with Torrent Variant Caller plug-in software. After variant detection, their effects were analyzed using the ANNOVAR software. The missense, nonsense, insertion/deletion, and splicing variants were selected from among the identified variants. Variants were further selected as less than 1% of (1) the 1000 genome database, (2) the 6500 exome variants, (3) the Human Genetic Variation Database (dataset for 1208 Japanese exome variants), and (4) the 333 in-house Japanese normal hearing controls by using our database software. Direct sequencing was utilized to confirm the selected variants.

The pathogenicity of the candidate variants was interpreted based on the standards and guidelines of the ACMG (American College of Medical Genetics) [23]. Co-segregation analysis was performed for each proband and their family members by using direct sequencing.

### 2.4. Mutagenesis 

Wild-type *MYO6* cDNA, fused with the flexi Halotag, was purchased from Kazusa Genome Technologies (pFN21A-*MYO6*, clone id hj00061, KGT, Chiba, Japan). The identified mutations were introduced by Infusion HD cloning (Takara-Bio, Shiga, Japan) according to the manufacturer’s protocol. Briefly, site-specific primers with the mutant variations were constructed (Appendix A) (Sigma-Aldrich, St. Louis, MO, USA) and PCR was performed for 9 mutations; c.374C > A, c.604A > G, c.614G > A, c.647A > T, c.1376G > A, c.1455T > A, c.2111G > A, c.2438G > C, and c.3746T > C, under the conditions of 35 cycles of 98 °C for 10 s, 58 °C for 30 s, and 72 °C for 8 min. Template DNA was then digested by DPN1 and the PCR product was purified with a PCR Purification kit (Qiagen, Hilden, Germany). PCR products were then re-circularized with the Infusion HD enzyme and transformed into TOP10 competent cells (Thermofisher Scientific, Waltham, MA, USA). Plasmid DNA was amplified and confirmed to possess the targeted mutations by Sanger sequencing (Appendix A). Plasmid DNA was also confirmed to not contain off-target mutations by Sanger sequencing.

### 2.5. Transfection

LLC-PK1-CL4 porcine kidney epithelial cells (CL4 cells) were acquired from the James Bartles lab [24]. The day before transfection, CL4 cells were seeded at 50% confluency in DMEM with 10% FBS (Thermofisher Scientific, Waltham, MA, USA) and grown overnight at 37 °C in a 5% CO_2_ incubator in a 12-well plate on 15-mm glass coverslips (Matsunami Glass, Osaka, Japan) precoated with 2% gelatin. Wild-type and mutant *MYO6* plasmid constructs were co-transfected with GFP-espin1 (obtained from the James Bartles lab) [24] with Lipofectamine 3000 (Thermofisher Scientific, Waltham, MA, USA) according to the manufacturer’s protocol. Briefly, 1 μg of DNA for each construct was prediluted in OptiMEMmedium (Thermofisher Scientific, Waltham, MA, USA) and combined with a pre-diluted Lipofectamine reagent, incubated at RT for 5 min, and then added to the cells. After 36 h, cells were checked for GFP expression by epifluorescent microscopy and fixed for immunocytochemistry.

### 2.6. Immunocytochemistry

Cells on coverslips were fixed in 4% paraformaldehyde in PBS for 10 min and then washed three times briefly in PBS, permeabilized for 20 min at room temperature with 0.1% Triton X-100 (Sigma-Aldrich, St. Louis, MO, USA) in PBS, washed in PBS 3 times, and then blocked with 3% goat serum (Thermofisher Scientific, Waltham, MA, USA) in PBS for 20 min. Cells were then incubated for 1 h at room temperature with a 1:200 dilution of the rabbit anti-Halotag antibody (Promega, Madison, WI, USA). Cells were washed three times with PBS, and then incubated for 1 h at room temperature with a 1:200 dilution of Alexa Fluor 546 goat anti-rabbit antibody and a 1/1000 dilution of DAPI. Cells were again washed in PBS 3 times, then the coverslips were mounted onto a slide glass with a Pro-long Gold Antifade mounting medium (Thermofisher Scientific, Waltham, MA, USA). Images were taken with an Olympus Fluoview FV-10i confocal microscope (Olympus, Okaya, Japan). The lengths of the microvilli with incorporated GFP-espin1 were measured with ImageJ software (NIH, Bethesda, MD, USA) and statistical significance assessed by Student’s *t*-test.

## 3. Results

### 3.1. Identified Mutations and Pathogenicity Interpretation 

A total of 27 possibly disease-causing *MYO6* candidate variants (six nonsense variants, five frameshift variants, seven splicing variants, one non-frameshift deletion, and eight missense variants) were identified from 32 AD inherited families and a single proband known to be affected with unknown inheritance in a heterozygous states (Table 1). Among these 33 *MYO6*-associated hearing loss families, seven families were reported in our previous paper. Thus, the prevalence of *MYO6*-associated hearing loss was 2.40% (32/1336) of Japanese autosomal dominant sensorineural hearing loss (ADSNHL) patients and 0.41% (33/8074) of all inheritance modes of Japanese hearing loss patients. All identified variants were confirmed by Sanger sequencing and the pathogenicity of these variants was interpreted in accordance with the ACMG guidelines [23]. As a result, three variants were classified as “pathogenic”, 16 as “likely pathogenic”, and 8 as “uncertain significance”. In cases where family member DNA samples were also obtained, segregation analyses were performed (Figure 1).

Among the “pathogenic” variants, p.R1166X was previously reported as a cause of autosomal recessive sensorineural hearing loss (ARSNHL) and ADSNHL [11,20]. In this study, this p.R1166X variant was identified from four cases with AD inheritance and a single proband known to be affected with unknown inheritance. 

### 3.2. Clinical Characteristics of MYO6-Associated Hearing Loss 

The clinical features of 38 cases from 33 families with *MYO6* variants are summarized in Table 1.

The average onset age is 24.0 years (range, 0 to 70 years). Among the 24 cases in which the age of onset was clarified, 22 cases occurred at the age of 40 or younger (Figure 2). Most of the cases suffered from mild-to-moderate hearing loss. However, three cases suffered from severe hearing loss and three cases suffered from profound hearing loss. Of 29 cases, an anamnestic evaluation indicated that 22 cases (75%) showed a progression of hearing loss. 

The pure-tone averages of the patients with *MYO6*-associated hearing loss and their ages when their hearing was tested are plotted in Figure 3A. The estimated hearing deterioration speed was 0.56 dB overall per year. However, when the analysis is restricted to over 40-year-old patients, the deterioration speed was increased to 1.07 dB per year. The averaged hearing thresholds for each frequency for every 20 years is indicated in Figure 3B. Patients who were 0−40 years old showed mild-to-moderate mid-frequency hearing loss; however, in the 41- to 60 age group, hearing deteriorated mainly in the higher frequencies and showed moderate high frequency-associated hearing loss. Furthermore, regarding the 61−80 age group, hearing deteriorated in all frequencies and showed a severe flat-type hearing loss. Most patients did not experience vertigo, with only five cases reporting episodes. 

As shown in Appendix A, there were 11 cases with multiple hearing test results, showing the actual progression of hearing loss.

### 3.3. In Vitro Analysis of the Identified MYO6 Variants 

To clarify the functional effects of the *MYO6* candidate missense variants identified in this study and the previous studies [20], we constructed a wild-type *MYO6* plasmid and incorporated the following identified variants by site-directed mutagenesis: p.P125H, p.N202D, p.R205Q, p.E216V, p.G459D, p.N485K, p.G704D, p.R813P, and p.F1249S (Figure 4).

We then transfected them into CL4 cells with a GFP-espin1 plasmid [25]. The microvilli of the CL4 cells were elongated by espin overexpression and provided a useful structure with which to investigate stereocilia-related proteins.

As a result, when we transfected only the GFP-espin1 plasmid or GFP-espin1 with the wild-type *MYO6* plasmid, cells exhibited the formation of long microvilli, indicating that the overexpression of *MYO6* itself does not affect microvilli formation (Figure 4B). However, when we incorporated GFP-espin1 with most candidate variants of *MYO6*, only shorter microvilli were formed (Figure 4B, Appendix A). In addition, all mutated myosin 6 proteins were accumulated at the base of the microvilli. The lengths of the microvilli formed in transfected cells were measured and statistical analysis was performed (Figure 4C). All mutations, except p.P125H, showed significantly shorter microvilli compared to wild-type *MYO6*.

## 4. Discussion 

This study expanded our previous study [20] and identified 33 families with candidate *MYO6* gene variants (Table 1) and, using the largest number (*n* = 8074) of HL patients with *MYO6*-associated hearing loss to date, clarified the precise data regarding frequency, mutation spectrum, and clinical characteristics of associated hearing loss. 

The frequency of *MYO6*-associated hearing loss was 2.40% in the Japanese AD hearing loss patients. This frequency is compatible with our previous study (2.6%; 7/266) [20] and is lower than the most frequent ADSNHL gene (6.6% for *KCNQ4* [26]) but comparable with other ADSNHL genes, such as 3.2% for *TECTA* [27], 2.5% for *POU4F3* [28], and 2.5% for *WFS1* [29]. Therefore, *MYO6* is considered an important causative gene for Japanese patients with ADSNHL. However, this is not a population-specific phenomenon as Seco et al. recently also reported that the *MYO6* gene is a frequent cause of AD hearing loss in the population in the Netherlands [30].

It is noted that, as shown in Figure 2, *MYO6*-associated hearing loss is mainly found in late-onset mild hearing loss cases, who are easily overlooked. MPS is useful in identifying the causative gene mutations in such juvenile-onset mild-to-moderate hearing loss cases. Indeed, herein, we report that *MYO6*-associated hearing loss is characterized by juvenile-onset mild to moderate hearing loss up to age 40. Thirty out of thirty-three cases in this study also showed juvenile-onset before the age of 40, which is consistent with our previous reports (Figure 2). We previously reported that the onset of hearing loss in *MYO6*-associated cases could be detected between the ages of 5 and 50, with the hearing deteriorating to profound hearing loss by the 6th decade [20]. We have shown that, in half of all individuals, hearing loss first appeared in the lower and mid frequencies and progressed to flat or sloping severe-to-profound hearing loss [20]. 

Most *MYO6*-associated hearing loss reported in previous papers showed progressive hearing loss [20,31,32]. Our results were consistent with such previous reports and most patients in this study also showed progressive hearing loss. It is noteworthy that the hearing thresholds were maintained until the age of 40, but rapidly deteriorated after that, worsening to severe hearing loss after the age of 60 (Figure 3B). In this study, multiple audiograms were obtained in 11 cases showing progressive hearing loss, providing supporting evidence of the progressive nature of the hearing loss due to *MYO6* gene variants (Appendix A). When the analysis was restricted to patients over 40 years of age, the deterioration speed increased to 1.07 dB per year. The reason for such an accelerated rate of deterioration for patients over 40 years old patients is unknown, but it may be related to the type of variants found in this study. Most of the variants found are non-functional variants (nonsense, frame shift, or splicing variants), and, therefore, it is plausible that any dysfunctions may be due to haploinsufficiency. If non-functional or unstable proteins were produced by these mutant alleles, such conditions may become more critical with age.

*MYO6* is known to cause ARSNHL (DFNB37) [11] and ADSNHL (DFNA22) [10]. The p.R1166X variant is an interesting one as it was first reported as a cause of AR hearing loss (DFNB 37) [11], but was later reported in three independent ADSNHL (DFNA22) families [20]. An in vitro functional study of the p.R1166X variant using RPE cells showed the pathogenicity of this mutation, indicating that this variant may cause cellular dysfunction [33].

In the present large cohort, the p.R1166X variant was found in four ADSNHL families and a single proband known to be affected with unknown inheritance, but unfortunately, we could not obtain audiograms from the parents in the sporadic family.

A possible explanation of how the p.R1166X variant causes both recessive and dominant inherited hearing loss involves autosomal semi-dominant inheritance. Indeed, an autosomal semi-dominant inheritance mode has been previously reported in *MYO6*-related hearing loss [34]. In the reported family, the family member with homozygotes for a splice mutation, c.897G>T, showed a more severe phenotype with early-onset, compared with the family members with heterozygotes with late-onset progressive hearing loss [34].

It should be noted that the phenotype of the DFNB 37 family [11] is based on anamnestic evaluation. Based on our clinical data, the phenotype caused by the p.R1166X variant is mild, and there is a risk that hearing loss could be overlooked unless audiometric evaluation is performed, or the parents are too young to develop hearing loss. 

Since the p.R1166X variant has been reported in other populations with different origins [11,20,35], this mutation may be present due to a hot spot rather than a common ancestor phenomenon.

In vitro analysis showed that the espin1-induced microvilli of CL4 cells become shorter when patient-identified missense *MYO6* variants were overexpressed. This observation could provide supportive evidence of the pathogenicity of the identified *MYO6* mutations. A mutant mouse model with a homozygous *MYO6* gene splicing mutation showed fused and/or branched stereocilia [10]. It is possible to cause such morphological abnormalities as a result of the inhibition of the stereocilia extension. Structural protein interactions at the base of stereocilia requires proper myosin 6 function. *RDX*, *CLIC5*, *TPRN*, and *PTPRQ* are the components which interact with *MYO6* to create protein complexes regulating stereocilia formation, maintenance, and mechano-transduction of the sound [36,37]. Mutations in the *MYO6* motor domain have been shown to inhibit *MYO6* localization in hair cells [36]. Interestingly, we found that mutant myosin 6 proteins were also localized in the basolateral region of the espin1-induced microvilli. These results indicate that the ability of the mutant myosin 6 to move to the basolateral end of microvilli was not disrupted. It has also been shown that myosin 6 is endogenously expressed in CL4 cells [38] and is estimated to have the ability to form the basal complex with myosin 6 proteins. When CLIC5 was overexpressed in CL4 cells, it directly interacted with TPRN and endogenous ezrin protein [36], indicating the proper functioning of this basal complex. From these observations and our results, mutations in *MYO6* may still allow myosin 6 proteins to maintain their motor function, as they are located in the basal region of microvilli, but disrupt complex formation or binding to other component proteins. In this paper, most of the candidate mutations were found to result in shorter microvilli; however, we could not identify any correlation between the length of the microvilli and the severity of hearing loss. Thus, further functional studies are required to be elucidated this phenotype–genotype correlation and molecular function. 

## 5. Conclusions

In summary, using a large cohort, we could obtain a more complete pictures of the mutational spectrum, frequency of mutations, and characteristic clinical features of patients suffering from hearing loss caused by *MYO6* variants. The better prediction of the hearing loss progression speed allows us to provide more appropriate intervention for the patients, such as hearing aids or cochlear implants, with optimal timing. Careful attention should be paid to the patients with late-onset mild-to-moderate progressive hearing loss as there is the potential for marked hearing deterioration to occurs after the age of 40.

## Figures and Tables

**Figure 1 genes-11-00273-f001:**
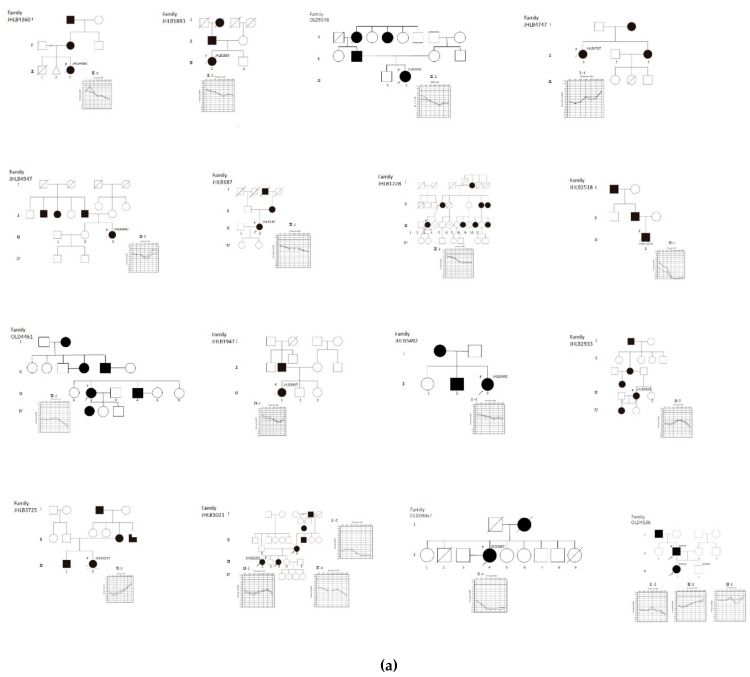
Pedigrees and audiograms of patients with *MYO6*-associated hearing loss. Arrowheads indicate family members receiving genetic testing, and arrowheads accompanying a “P” indicate the probands. (**a**) Filled symbols indicate affected individuals, and open symbols indicate unaffected individuals. (**b**) Pedigrees and audiograms of patients with *MYO6*-associated hearing loss.

**Figure 2 genes-11-00273-f002:**
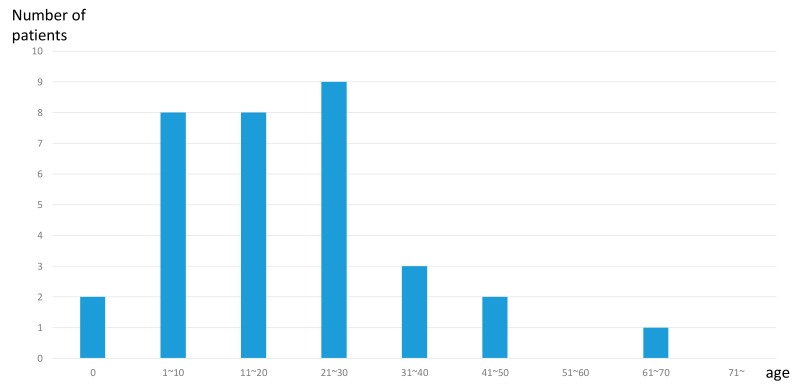
Distribution of the onset age of *MYO6*-associated hearing loss cases. Histogram of 33 cases segregated into 10-year intervals or congenital.

**Figure 3 genes-11-00273-f003:**
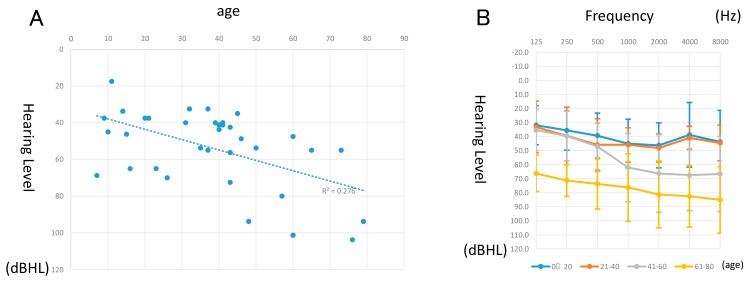
Hearing loss deterioration in patients with *MYO6* mutations. (**A**) Correlation between age and hearing. Pure-tone averages of each patient were plotted by dB hearing level (HL) and age at the time of the hearing test. (**B**) Average hearing threshold for each 20-year age group for patients with *MYO6* mutations. A total of 8 cases were indicated in the 0−20, 11 in the 21−40, 12 in the 41−60, and 4 in the 60 or older.

**Figure 4 genes-11-00273-f004:**
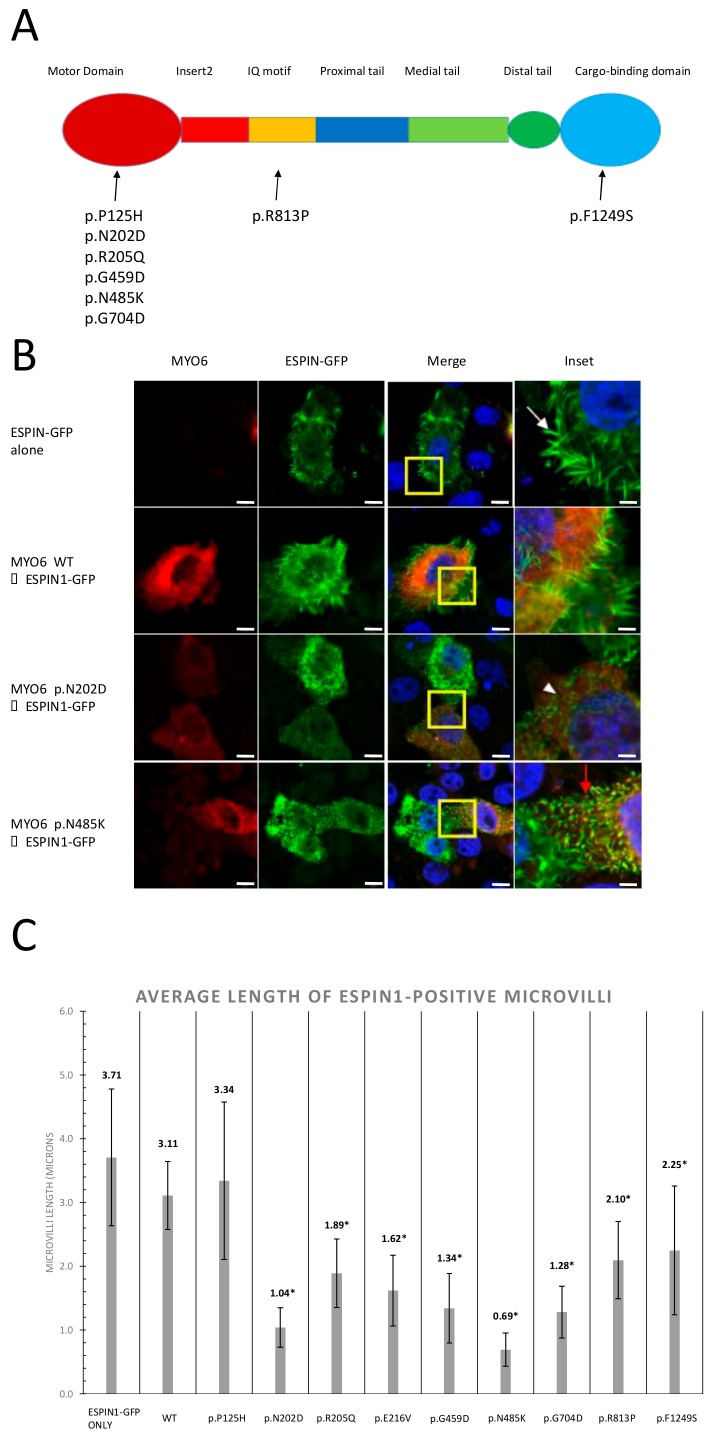
Mutations in *MYO6* resulted in shorter microvilli. (**A**) Locations of targeted mutations in the myosin 6 domain structure as indicated. Most mutations were located in the motor domain. (**B**) Confocal images of immunocytochemistry of CL4 epithelial cells transfected with GFP-espin1 alone or co-transfected with Halo-tagged *MYO6* constructs as indicated. The yellow boxes are magnified as the insets. *MYO6* overexpression was detected using an anti-Halotag antibody. GFP-espin1 induced long microvilli (white arrow), but these were shorter in cells with mutant myosin 6 (white arrowhead). *MYO6* expression can be detected at the base of each microvilli (red arrow). Scale bars = 10 µm. **(C)** Graph indicating the average length of GFP-espin1-positive microvilli. Microvilli length of cells with each mutation were measured in ImageJ (µm). For each sample, 50−70 microvilli were measured for length. Asterisks indicates statistically significant differences compared to the WT (*p* < 0.001).

**Table 1 genes-11-00273-t001:** Possibly disease-causing *MYO6* candidate variants.

Family	Subject	Exon	Base Change	AA Change	Onset (y.o)	Age (y.o)	Progression	Vertigo	Audiogram (dB)	SIFT	PP2HV	LRT	Mut Taser	Mut Assesor	CADD Phred	ACMG Criteria	Heredity	
JHLB4360	Ⅲ-3	3	c.187_187del	p.C63fs	under 10	9	NA	NA	NA							likely pathogenic	AD	
JHLB1893	Ⅲ-1	4	c.201delT	p.Y67fs	20	31	+	+	40							likely pathogenic	AD	
OLD5048	Ⅲ-2	4	c.238C > T	p.R80X	18	37	+	–	55			D	A		39	pathogenic	AD	
JHLB4747	Ⅱ-1	5	c.374C > A	p.P125H	30	60	+	–	47.5	D	D	D	D	H	25.9	uncertain significance	AD	
JHLB4947	Ⅲ-3	6	c.429_431del	p.143_144del	20	40	+	–	41.25							likely pathogenic	AD	
JHLB187	Ⅲ-2	7	c.553 + 1G > T	splicing aberrant	20	41	+	–	40				D		24.9	likely pathogenic	AD	
JHLB1228	Ⅲ-2	8	c.577delG	p.G193fs	NA	45	–	–	35							likely pathogenic	AD	
JHLB2518	Ⅲ-1	8	c.604A > G	p.N202D	5	60	+	–	101.25	D	D	D	D	H	24.2	uncertain significance	AD	
OLD4461	Ⅲ-2	8	c.614G > A	p.R205Q	50	65	+	–	55							uncertain significance	AD	＊
JHLB1947	Ⅲ-1	10	c.863_866del	p.D288fs	8	20	–	–	37.5							pathogenic	AD	
JHLB3492	Ⅱ-3	10	c.863_866del	p.D288fs	NA	NA	NA	NA	NA							pathogenic	NA	
JHLB2933	Ⅲ-2	12	c.1079 – 2A > G	splicing aberrant	27	39	+	+	40				D		24.3	likely pathogenic	AD	
JHLB3725	Ⅲ-2	13	c.1376G > A	p.G459D	12	23	–	–	65	D	D	D	D	H	25.4	uncertain significance	AD	
JHLB1021	Ⅲ-1	14	c.1455T > A	p.N485K	27	50	+	–	53.75	D	D	N	D	H	23.2	uncertain significance	AD	
JHLB1021	Ⅱ-2	14	c.1455T > A	p.N485K	20s	79	+	–	NA	D	D	N	D	H	23.2	uncertain significance	AD	
JHLB1021	Ⅲ-3	14	c.1455T > A	p.N485K	elementary school	46	NA	–	NA	D	D	N	D	H	23.2	uncertain significance	AD	
OLD3667	Ⅱ-4	19	c.1975C > T	p.R659X	50	76	+	+	103.8			D	A		48	likely pathogenic	AD	＊
OLD4536	Ⅱ-2	19	c.1975C > T	p.R659X	28	43	+	–	72.5						48	likely pathogenic	AD	＊
OLD4536	Ⅲ-1	19	c.1975C > T	p.R659X	9	9	+	–	37.5						48	likely pathogenic	AD	＊
OLD4536	Ⅲ-2	19	c.1975C > T	p.R659X	pre-critical	11	NA	–	17.5						48	likely pathogenic	AD	＊
JHLB4574	Ⅲ-2	20	c.2077 + 3A > G	splicing aberrant	70	73	+	–	55							likely pathogenic	AD	
OLD4674	Ⅲ-1	21	c.2111G > A	p.G704D	6	14	–	–	33.75	D	D	D	D	H	24.4	uncertain significance	AD	
JHLB940	Ⅳ-3	22	c.2209 – 2A > G	splicing aberrant	20	21	–	+	37.5				D		25	likely pathogenic	AD	
JHLB433	Ⅲ-1	23	c.2287 – 2A > G	splicing aberrant	6	10	NA	–	45				D		24.8	likely pathogenic	AD	
JHLB3986	Ⅲ-1	23	c.2416 + 5G > A	splicing aberrant	20	57	+	+	80							likely pathogenic	AD	
JHLB1589	Ⅳ-1	24	c.2438G > C	p.R813P	0	7	–	–	68.75	D	D	D	D	M	27.4	uncertain significance	AD	
OLD4362	Ⅲ-3	25	c.2563_2564insT	p.E855fs	6	26	+	–	70							likely pathogenic	AD	
OLD3510	Ⅲ-3	26	c.2839C > T	p.R947X	24	35	+	–	53.75			D	A		38	likely pathogenic	AD	
OLD2267	Ⅲ-2	28	c.2998C > T	p.Q1000X	30	43	NA	NA	56.25			D	A		42	likely pathogenic	AD	
OLD2267	Ⅱ-2	28	c.2998C > T	p.Q1000X	NA	60s	NA	NA	NA			D	A		42	likely pathogenic	AD	
JHLB1235	Ⅱ-1	32	c.3361A > T	p.K1121X	26	41	+	–	41.25			D	A		50	likely pathogenic	AD	＊
JHLB530	Ⅲ-4	34	c.3496C > T	p.R1166X	29	40	NA	–	43.75			D	A		47	pathogenic	AD	＊
JHLB315	Ⅳ-2	34	c.3496C > T	p.R1166X	35	37	+	–	32.5			D	A		47	pathogenic	AD	＊
JHLB193	Ⅲ-1	34	c.3496C > T	p.R1166X	31	32	NA	NA	32.5			D	A		47	pathogenic	AD	＊
JHLB3296	Ⅱ-2	34	c.3496C > T	p.R1166X	NA	36	+	–	65			D	A		47	pathogenic	sporadic	
JHLB3050	Ⅱ-2	34	c.3496C > T	p.R1166X	30	48	+	–	93.75			D	A		47	pathogenic	AD	
OLD2149	Ⅲ-2	35	c.3659 – 2A > G	splicing aberrant	39	43	+	–	42.5				D		23.4	likely pathogenic	AD	
JHLB3236	Ⅳ-3	35	c.3746T > C	p.F1249S	0	15	–	–	46.25	D	D	D	D	M	25.6	uncertain significance	AD	

AAChange, amino acid change; PP2HV, PolyPhen2 HumVar; MutTaster, MutationTaster; MutAssessor, MutationAssesor, ACMG, American College of Medical Genetics; D, deleterious or probably damaging; A, disease-causing-automatic; M, medium; H, high; N, neutral; AD, autosomal dominant; NA, data was not available. ＊: previously reported case [20].

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
