# Peer review of "Clinical Characteristics and In Vitro Analysis of MYO6 Variants Causing Late-onset Progressive Hearing Loss"

_genes, 2020, doi:10.3390/genes11030273_

Round 1
Reviewer 1 Report
Oka and colleagues performed massively parallel sequencing in > 8000 Japanese families with hearing loss. The authors identified 27 MYO6 variants in 33 families, including 22 novel variants which have never been documented before. The authors then delineated the clinical features in their patients: most patients showed juvenile onset progressive hearing loss with the hearing deteriorating markedly after 40 years of age. The author also performed functional experiments by transfecting 9 missense variants into CL4 cells, and confirmed the pathogenicity of these variants.
In general, this paper is well-written and well-organized. Both the genotyping and functional studies were thoroughly conducted. The major strength of this paper lies in the comprehensive clinical data retrieved from a large cohort. The findings of this article, by clearly delineating the audiological features of MYO6-related hearing loss, are also clinically relevant.
Minor comment:
1. The finding about the accelerated deterioration of hearing loss after 40 y is interesting. Do the authors have an explanation for this? It would be interesting to address this point in the Discussion section.
Author Response
<Reviewer 1>
- The finding about the accelerated deterioration of hearing loss after 40 y is interesting. Do the authors have an explanation for this? It would be interesting to address this point in the Discussion section.
This is a good but difficult question, because there is no direct evidence to explain it at this moment. But it may be related to the type of mutation found in this study. We added the speculation stated below in the Discussion section.
“The reason for such accelerated deterioration over 40 years old is unknown, but it may be related to the type of variants found in this study. Most of the variants found are non-functional variants (nonsense, frame shift, or splicing variants), and therefore it is plausible that dysfunction maybe due tohaploinsufficiency. If non-functional or unstable proteins were produced by these mutant alleles, such conditions may become critical with age.”
Reviewer 2 Report
This is an interesting paper on MYO6 in the Japanese deaf. The functional studies add greatly to the manuscript. While generally well-written, there are a number of grammatical errors, e.g. lines 37 (sequencing, not sequencer), lines 42-43 (...year; when restricted to changes after 40 years of age, the ), lines 43-45, (..for pathogenicity, variants identified..), line 63 (..is located on chromosome...), line 271 (.. become severe), line 277 (variant (singular)), line 296 (...patient-identified), line 304-305 (not sure the best way to fix), 305-306 (...results indicate that...); there may be more instances throughout the manuscript.
The 50% mentioned in line 55 is for the United States. It probably includes all industrialized nations, of which Japan is one, but this should be noted.
line 121. The segregation analysis mentioned is actually co-segregation analysis of the phenotype and the variant.
lines 165, 166 and 247. First, the authors state that there are 31 AD families and 1 sporadic, then that there are 33 MYO6 associated families and then finally that there are 34. I count 33 pedigrees.
line 166, 281. Unless they are sure that the lone affected individual is deaf as the result of a new mutation, this should be referred to as a single known affected with unknown inheritance.
line 323 seems to just drop off: ...age of 4
Author Response
<Reviewer 2>
This is an interesting paper on MYO6 in the Japanese deaf. The functional studies add greatly to the manuscript. While generally well-written, there are a number of grammatical errors, e.g. lines 37 (sequencing, not sequencer), lines 42-43 (...year; when restricted to changes after 40 years of age, the ), lines 43-45, (..for pathogenicity, variants identified..), line 63 (..is located onchromosome...), line 271 (.. become severe), line 277 (variant (singular)), line 296 (...patient-identified), line 304-305 (not sure the best way to fix), 305-306 (...results indicate that...); there may be more instances throughout the manuscript.
> Thank you for pointing out these grammatical errors. They have been corrected.
The 50% mentioned in line 55 is for the United States. It probably includes all industrialized nations, of which Japan is one, but this should be noted.
> The following sentence was added.
> Genetic etiology is the most common cause of sensorineural hearing loss (SNHL), with over 50% of congenital hearing loss attributed to genetic causes in the US [1], and most likely similar in all individualized nations.
line 121. The segregation analysis mentioned is actually co-segregation analysis of the phenotype and the variant.
> The sentence was corrected as follows.
> Co-segregation analysis was performed on each proband and their family members by using direct sequencing.
lines 165, 166 and 247. First, the authors state that there are 31 AD families and 1 sporadic, then that there are 33 MYO6 associated families and then finally that there are 34. I count 33 pedigrees.
> Sorry for confusion. We re-checked the clinical data, and found HL1587 and JHLB4360 in Table 1 is identical family. Therefore, 33 is the correct number. We corrected the sentence and Table.
line 166, 281. Unless they are sure that the lone affected individual is deaf as the result of a new mutation, this should be referred to as a single known affected with unknown inheritance.
> According the reviewer’s comment, we used the term “a single known affected with unknown”.
line 323 seems to just drop off: ...age of 4
> Thank you for pointing out this. We corrected to “age of 40.”.